# Transcriptomic Changes Following Induced De-Masculinisation of Australian Red Claw Crayfish *Cherax quadricarinatus*

**DOI:** 10.3390/ijms24043292

**Published:** 2023-02-07

**Authors:** Grace Smith, Susan Glendinning, Tomer Ventura

**Affiliations:** 1Centre for Bioinnovation, University of the Sunshine Coast, Maroochydore, QLD 4556, Australia; 2School of Science, Technology and Engineering, University of the Sunshine Coast, Maroochydore, QLD 4556, Australia

**Keywords:** androgenic gland, insulin endocrinology, insulin-like peptides, monosex technology, red claw crayfish, reproduction, transcriptomic library

## Abstract

The Australian red claw crayfish *Cherax quadricarinatus*, an emerging species within the freshwater aquaculture trade, is not only an ideal species for commercial production due to its high fecundity, fast growth, and physiological robustness but also notoriously invasive. Investigating the reproductive axis of this species has been of great interest to farmers, geneticists, and conservationists alike for many decades; however, aside from the characterisation of the key masculinising insulin-like androgenic gland hormone (IAG) produced by the male-specific androgenic gland (AG), little remains known about this system and the downstream signalling cascade involved. This investigation used RNA interference to silence *IAG* in adult intersex *C. quadricarinatus* (*Cq-IAG*), known to be functionally male but genotypically female, successfully inducing sexual redifferentiation in all individuals. To investigate the downstream effects of *Cq-IAG* knockdown, a comprehensive transcriptomic library was constructed, comprised of three tissues within the male reproductive axis. Several factors known to be involved in the IAG signal transduction pathway, including a receptor, binding factor, and additional insulin-like peptide, were found to not be differentially expressed in response to *Cq-IAG* silencing, suggesting that the phenotypic changes observed may have occurred through post-transcriptional modifications. Many downstream factors displayed differential expression on a transcriptomic level, most notably related to stress, cell repair, apoptosis, and cell proliferation. These results suggest that IAG is required for sperm maturation, with necrosis of arrested tissue occurring in its absence. These results and the construction of a transcriptomic library for this species will inform future research involving reproductive pathways as well as biotechnological developments in this commercially and ecologically significant species.

## 1. Introduction

### 1.1. Aquatic Invasions

Subsequent to the increase in global trade over the last century, international aquaculture and aquarium trade have been chiefly responsible for the mass translocation of invasive marine species, not limited to fish [1,2], gastropods [3], and crustaceans [4,5,6]. Crustaceans are some of the most successful marine invaders, accounting for over half of all invasive species in European freshwater catchments [5]. Decapod crustaceans, a diverse taxonomic order, have been extensively translocated by humans due to their numerous applications to cultural, recreational, and economic practices [7,8]. The farming of freshwater crayfish *Astacidea*, known as astaciculture, is widespread throughout America, Europe, China, and Australia. Although difficult to quantify, farmed production of red swamp crayfish *Procambarus clarkii* in China has far surpassed 2 million tons annually, making it the most prominent astaciculture species globally [9]. An emerging species within the astaciculture trade is the Australian red claw crayfish *Cherax quadricarinatus*, an ideal species for commercial production due to impressive physiological robustness and a simple non-planktonic life cycle compared to many decapod species [10]. They display high fecundity, with females spawning up to five times per year with over 1000 eggs per spawn, and exhibit exceptionally fast growth, reaching commercial size in under 9 months in optimal conditions [11]. Due to these traits, invasive populations of *C. quadricarinatus* are notoriously known for breeding prolifically and outcompeting native or endemic species [10]. *Cherax* species are farmed at a high density, making them especially susceptible to translocation [12]. As a result of their growing popularity amongst farmers, *C. quadricarinatus* species have been subject to mass translocation outside their native range in northern Australia, with populations now widespread throughout Australian catchments as well as internationally [6,13]. Understanding the neuroendocrine factors driving sexual differentiation and reproduction in *C. quadricarinatus,* therefore, has valuable applications to both increase farming capacity and reduce invasive potential.

### 1.2. Biotechnology Solutions

Monosex aquaculture, in which populations are either all-male or all-female, is a desirable practice in animal husbandry when a specific sex is larger, faster growing, or more attractive. All methods of generating monosex crustacean populations have thus far relied on an initial sex-reversal via endocrine manipulation, from female to male or vice versa, to create a neo-male or neo-female, which can be bred to produce all-male or all-female progeny [14]. In the giant freshwater prawn *Macrobrachium rosenbergii*, this patented method has been used to successfully generate commercial all-male populations, facilitating higher production over shorter time scales as a result of faster and more uniform growth [15]. Similar results have yet to be replicated in other decapod species, despite decades of research investigating the means of sexual plasticity and the mechanisms of sexual differentiation within crustaceans [16].

It has been suggested on multiple occasions that culturing all-male populations of *C. quadricarinatus* would improve yield and be economically advantageous as males grow larger and have a higher market value in comparison to females due to the eponymous red claw [17,18,19]. A prior study found that manually sorted all-male populations of *C. quadricarinatus* grew to be significantly larger than all-female or mixed-sex populations, reporting faster growth, increased yield, and more efficient feed conversion [18]. Manual sorting is obviously unsuitable for an industrial scale due to cost and time constraints; however, utilizing biotechnology to produce all-male populations on an industrial scale would be more economically advantageous to the freshwater aquaculture industry in addition to being a considerable milestone in neuroendocrine decapod research [19]. The development of this technology is primarily limited by a lack of understanding of the neuroendocrine pathways involved in regulating sexual differentiation. It is possible that other unknown factors within this axis could also yield beneficial results to aquaculture if they could be identified and manipulated.

*C. quadricarinatus* displays a uniquely high natural degree of sexual plasticity, with intersex individuals comprising up to 14 percent of both wild and cultured populations [20]. Intersex *C. quadricarinatus* possess both male and female external gonopores as well as non-paired male and female reproductive organs simultaneously and are genotypic females but functional males with viable spermatozoa and a previtellogenic ovary like that found in a juvenile female [21,22]. It has been suggested the high degree of sexual plasticity demonstrated by *C. quadricarinatus* provided an evolutionary fitness advantage, allowing excess production of females during periods of low reproductive output [23].

### 1.3. The Neuroendocrine Network

Contrary to the general pattern observed across vertebrates, sex steroids such as oestrogens and androgens are not responsible for sexual development in crustaceans; rather, this function is regulated by hormones within the insulin-like superfamily of peptides (ILPs), which are found widely throughout invertebrates and are, among other processes, fundamental to the mediation of primary and secondary sexual characteristics [24,25,26]. While chromosomal sex determination occurs at fertilization, shifts towards masculinity or femininity may be induced within the lifetime of an animal in many crustacean species due to the function of ILPs [27].

The androgenic gland (AG), first discovered in the blue swimming crab *Callinectes sapidus* in 1947, is the only known sex-specific gland in crustaceans [28,29]. Six decades after the initial discovery of the AG, the associated masculinizing insulin-like AG hormone (IAG) was discovered in *C. quadricarinatus* [30], and has since been identified in many other *Astacidea* species [16,27]. Almost universally across crustacean species, the AG and associated IAG are exclusively found in males and are responsible for the development of primary and secondary male sexual characteristics, including spermatogenesis and increased somatic growth rates [16].

Multiple studies on *C. quadricarinatus* using targeted gene silencing via RNA interference (RNAi) as well as AG ablation have demonstrated de-masculinization and the arrest of the male reproductive system in the absence of IAG, indicating that male gonad viability is reliant on this hormone [31,32]. In intersex *C. quadricarinatus*, IAG silencing also caused upregulation of the ovary, which is typically pre-vitellogenic in intersex individuals [31]. The inhibitory effect of IAG on female development has been demonstrated by AG implantation in females, inducing the suppression of female characteristics such as yolk production and the development of secondary masculine traits such as distinctive red calcareous patch on the claw, reduced aggression towards other females, and masculinization of the pleopods [33,34,35].

There are many complex neuroendocrine pathways in decapods, which regulate a multitude of physiological processes [28]. The X-organ–sinus gland neuroendocrine complex (XO–SG) residing in the eyestalk negatively regulates many of these processes, including the reproductive axis, via the crustacean hyperglycaemic hormone (CHH) superfamily of inhibitory neuropeptides [16,36]. The removal of these inhibitory peptides triggers cascading effects resulting in moulting, gonad maturation, and spawning [37]. The eyestalk–AG–testis axis (ES-AG-TS) is a complex neuroendocrine network involving many tissues, proteins, receptors, and feedback loops, many of which have not been fully explored (Figure 1). Eyestalk ablation in male *C. quadricarinatus* results in hypertrophy of the AG and increased spermatogenesis [28]. A prior study on intersex C. quadricarinatus reported hypertrophy of the AG following IAG silencing and suggested a feedback loop in which IAG indirectly moderates its own production [31]. The XO-SG complex also negatively regulates vitellogenin, subsequently meaning that it also has an inhibitory effect on the hepatopancreas, where the vitellogenin gene is synthesized [38].

### 1.4. Broader Insulin Endocrinology

While IAG is the cardinal hormone secreted by the AG, many other hormones and peptides are necessary for its synthesis, secretion, and functioning, including other insulin-like peptides (ILPs), insulin-like growth factor (IGF), membrane-anchored AG-specific factor (MAG), and tyrosine kinase insulin receptor (TKIR). Common throughout the entire animal kingdom, the insulin-like superfamily function as neuroendocrine hormones in invertebrates. ILPs, a subfamily of the insulin-like superfamily, signal primary and secondary sexual differentiation in crustaceans [24,25,26]. ILPs are thought to have evolved from a single ancestral insulin gene as they typically share a similar structure [24]. ILPs are encoded as a two-peptide chain structure, with an A-chain, B-chain, and connecting C-peptide [24,26]. To date, the three ILPs described in *C. quadricarinatus* are IAG, ILP1, and ILP2. In the eastern spiny lobster *Sagmariasus verreauxi,* ILP1 is expressed in the brain, antennal gland, gonads, and female hepatopancreas, while ILP2 is expressed in the brain and thoracic ganglia; however, their specific functions within the network are yet to be described [25].

The insulin-like growth factors (IGFs), another subfamily of the insulin-like superfamily, are structured similarly to ILPs but also contain a D-domain and E-domain after the A-chain [24]. IGFs are highly conserved throughout evolutionary history, functioning similarly in vertebrates and invertebrates to regulate cell differentiation and apoptosis among other functions [24]. Insulin-like growth factor binding proteins (IGFBPs) bind with IGFs as well as ILPs, acting as a bio-regulator for IAG [25]. IAG is pleiotropic, inducing signalling in multiple tissue types. IGFBP is suggested to bind to IAG in muscle, epithelial, and gonad cells, thereby regulating its expression throughout tissues [24]. *IAG* silencing in *Macrobrachium nipponense* has been shown to result in a reduction in *IGFBP* expression across the entire TS–AG–hepatopancreas axis, excluding the neuroendocrine tissues [39], which may indicate a maintained function in these tissues [25].

Insulin receptors are required for ILP signalling pathways; however, little is understood about these networks and insulin-like receptors have so far been poorly described [40]. IAG regulates downstream phosphorylation events through the tyrosine kinase insulin receptor (TKIR), a transmembrane receptor belonging to the tyrosine kinase superfamily [41]. The binding of IAG with the receptor triggers a cascade of phosphorylation events, ultimately resulting in spermatogenesis in the testis. Demonstrated in *S. verreauxi,* TKIR is also expressed in the antennal gland, indicating that this gland may also be involved in the AG endocrine axis [41].

Like IAG, MAG is an AG-specific protein, and was the first non-insulin-like AG transcript reported in crustaceans [42]. When fused to GFP, the expression pattern of MAG has been shown to change from localized around the nucleus in an ER-like pattern to scattered throughout the cell when the anchoring N-terminus of the protein was removed [42]. While the function of MAG within the AG has not yet been described, it is hypothesised to be involved in the processing and secretion of IAG.

### 1.5. Research Aims

Despite seven decades of research since the AG was first discovered, there are many downstream factors involved in the IAG signalling cascade that remain poorly understood or unknown entirely. Relatively little genomic and transcriptomic analysis has been conducted on *C. quadricarinatus* in comparison to other commercial decapod species such as *M. rosenbergii* [43] or *P. clarkii* [44], meaning that many aspects have never been explored or characterised. The future development of biotechnology solutions—for which there is a high demand from both farmers and conservationists—cannot progress without first developing a better understanding of the neuroendocrine network. This study aims to broaden our understanding of the function of IAG within the reproductive axis by the construction of a comprehensive transcriptomic library assembled from both control and *Cq-IAG*-silenced tissues.

## 2. Results

### 2.1. Anatomical Measurements and Observations

An effective sex switch was induced, indicated by the regeneration of feminised pleopods after moult in the *Cq-IAG*-silenced group (Figure 2a). In comparison with the control-injected animals, the pleopods of *Cq-IAG*-silenced animals had clearly been feminised, with the endopod being approximately twice the width of the exopod (Figure 2a,e). Upon dissection, all animals in the *Cq-IAG*-silenced group were found to have enlarged, yolk-filled, pigmented ovaries and depleted testis and sperm duct (Figure 2b–d). When compared to the control animals, the oocytes of *Cq-IAG*-silenced animals were clearly larger and actively producing yolk (Figure 2d). Oocytes were enlarged and bright yellow in colour, characteristic of what would be expected in reproductively active mature females. All ovaries in the control-injected group were pre-vitellogenic with low pigmentation, and entirely absent in one animal, as typically seen in juvenile females or untreated intersex animals (Figure 2h). The sperm ducts of *Cq-IAG*-silenced animals all appeared to be entirely depleted of sperm and in several cases partially or entirely necrotic, seen as melanised tissue (Figure 2c). Testes in control animals were actively producing sperm and sperm ducts appeared otherwise normal and healthy (Figure 2f,g). In comparison to the control group, the *Cq-IAG*-silenced group had a significantly lower testis mass (F (df residual = 9, df test = 1) = 16, *p* = 0.00311) with a mean percentage mass of 0.287 ± 0.0275 compared to the control groups mean percentage mass 0.479 ± 0.0408 (Figure 3A). The *Cq-IAG*-silenced group shower significantly greater ovary mass (F (df residual = 9, df test = 1) = 6.7, *p* = 0.0292) with a mean percentage mass of 0.335 ± 0.157 compared to the control groups’ mean percentage mass 0.0511 ± 0.0129 (Figure 3B).

Histological examination indicated that the *Cq-IAG*-silenced testis tissue was primarily composed of lobules filled with arrested spermatogonia that did not seem to be progressing to spermatocytes or mature spermatozoa (Figure 4). The cross section of the sperm duct was devoid of sperm fluid or sperm packets, with only remnants of degenerated cells present. Testicular lobules appeared to be arrested prior to secondary meiosis, with little to no indication of spermatocytes or spermatozoa, evidenced by the lack of cells with condensed cytoplasm. By comparison, in the control testis tissue, many different stages of cell maturity can be seen in adjacent lobules (Figure 4). Spermatogonia are present and proliferating, with spermatocytes and spermatozoa clearly abundant. Mature sperm packets are evident and the tissue appears healthy and active. The trend of active testis in the control group and inactive testis in the *Cq-IAG-*silenced group was consistent across all sampled tissues.

### 2.2. RNA-Seq Results

A total of 24 tissues were sequenced, returning 5.6–7.1 Gb total reads per library, with a minimum of 96.7% clean reads (based on minimum Phred score Q20 or above). A de novo transcriptome assembly of six FASTQ files (each from control and Cq-IAG-silenced eyestalk, testis, and AG) resulted in 67,582 transcripts. The 24 tissue FASTQ files are available through NCBI Sequence Read Archive under accession number PRJNA862554 and relative expression data for all 24 samples are available through CrustyBase [45]. A principal component analysis (PCA) was used to visualise clustering of the top 500 expressed transcripts between tissue replicates (Figure 5). The expression of the top 500 transcripts comprised over 86% of total transcript expression across all tissues. The top 500 expressed transcripts across the three tissues were plotted against the two main principal components, showing clear clustering of tissues, with the eyestalk tissues being more removed than the testis and AG tissues. There is no notable clustering of treatment groups (Figure 5), indicating that there was no significant differentiation in the top 500 expressed transcripts between the control and *Cq-IAG*-silenced groups.

Transcript analysis confirmed *Cq-IAG* expression was successfully knocked down in the AG (fold change 6.75, FDR *p* value < 0.0001) and was not significantly expressed in the other tissues (RPKM < 1). Of the other genes of interest previously-described, MAG (JX446634.1) was unable to be identified in the assembly, while ILP1 (KP006644.1), TKIR (identified via tblastn of XM_045768873.1 using *C. quadricarinatus* databases on CrustyBase.org), and IGFBP (KC952011.1) were not differentially expressed in any tissue examined (fold change between 2 and −2).

From the de novo assembled transcriptome, differential expression between the control and *Cq-IAG*-silenced groups was determined for each tissue. Twenty-seven transcripts were differentially expressed in the AG and 63 transcripts were differentially expressed in the testis; however, no transcripts were found to be differentially expressed in the eyestalk. Of the total 90 differentially expressed transcripts, 26 transcripts in the AG and 34 transcripts in the testis were upregulated and one transcript in the AG and 29 transcripts in the testis were downregulated (Figure 6). In the AG, the only downregulated transcript was *Cq-IAG,* with all others being upregulated.

Of the 90 differentially expressed transcripts, 30 were provided annotation, while the remaining 60 were not found to have any significant match against NCBI databases (e-value > 1 × 10^−20^). Two transcripts were characterised as putative *C. quadricarinatus* transcripts, those being *Cq-IAG* and glutathione peroxidase 3 (*GPx3*). Other transcripts were annotated to the closest protein match in other species, in most cases the red swamp crayfish *P. clarkii* or the American lobster *Homarus americanus*. Functions including regulation of apoptosis, cell growth, and neural signalling were predicted for these transcripts (Figure 7).

## 3. Discussion

### 3.1. Understanding the Axis

Despite the relevance of *C. quadricarinatus* as both an economically and ecologically important species, little is understood regarding the mechanisms of sexual differentiation in this species. With the market demand for monosex populations of *C. quadricarinatus* increasing and the importance of this technology highlighted by multiple studies [17,18,19], the study of the reproductive axis and IAG signal transduction pathway is of high importance. This project used gene silencing followed by transcriptomic analysis of differentially expressed transcripts across the male reproductive axis to investigate the extent of an association between selected male tissues and the proteins produced within them. The abnormal physiological observations of the ovaries and testes indicated that successful sexual redifferentiation had occurred, while transcriptomic analysis demonstrated cascading effects throughout the male reproductive axis following the silencing of *Cq-IAG* in intersex individuals. *Cq-IAG*-silenced adults displayed clear ovarian upregulation, with the ovaries becoming enlarged and vitellogenic, as well as the arrest of spermatogenesis, with the testes non-functional or necrotic. A large variation in the effect of *Cq-IAG* silencing on the gonadosomatic ovary index was observed (indicated by a higher standard error), compared to a very consistent effect on the testis index. This may be due to indirect signalling to the ovary resulting in greater variation. Histology of the testis and associated convoluted sperm duct indicated the arrest of spermatogenesis following *Cq-IAG*-silencing, with cells proliferating but not progressing to mature spermatozoa (Figure 5). These findings support a role for IAG in sperm maturation, with spermatogenesis arrested in its absence, building on previous literature reporting the upregulation of the ovary and apoptosis of the testis in juvenile *C. quadricarinatus* after *Cq-IAG* knockdown (Rosen et al., 2010).

While clear physiological abnormalities were observed in the testis and AG between the control and experimental groups at both a phenotypic and transcriptomic level, intriguingly, no such difference was observed upstream in the eyestalk. As inhibitory CHH peptides produced within the eyestalk are known to regulate the reproductive axis, it is interesting that there was no evidence of a feedback loop in this system, despite the cascading downstream effects. It has been proposed that IAG regulates its own expression via communication with the eyestalk [31]; however, this feedback appears to operate at a post-transcriptional level, evidenced by the lack of differential transcript expression in the eyestalk. Feedback may occur via micro-RNA pathways, or alternatively via post-transcriptional pathways. Both options should be explored in future studies, focusing on the phosphorylation of proteins in the testis. Translational or post-translational regulation is further supported by the fact that despite a clear phenotypic change, previously described proteins known to be crucial to the functioning of IAG within the reproductive system (ILP1, TKIR, and IGFBP) did not show differential expression in any tissue. In the case of TKIR, this may be due to the receptor remaining present but simply not being active in the absence of IAG. In *M. nipponense*, knockdown of *IAG* resulted in the downregulation of *IGFBP* across testis and AG tissues [39], while this was not the case in our study, perhaps due to IGFBP binding other ILPs in addition to IAG [25].

### 3.2. Differential Expression in the AG

Under a stringent fold change of >5, ninety transcripts were differentially expressed across all tissues in response to *Cq-IAG* silencing; zero in the eyestalk, 27 in the AG, and 63 in the testis. The only downregulated transcript in the AG was *Cq-IAG,* indicating the successful knockdown of the silencing target. Of the remaining 26 upregulated transcripts, 9 were provided annotation, four to known *Astacidea* proteins, and the remainder to hypothetical or uncharacterised *Decapoda* proteins.

A progranulin-like protein was upregulated in the AG, the function of which appears highly conserved between vertebrates and the few invertebrates in which it has been studied (namely, platyhelminths), being a growth factor involved in wound healing and inflammation [46,47]. This may indicate hypertrophy of the AG, as is usually evident following *Cq-IAG* silencing in decapods [16], or could be resulting from necrotic sperm duct tissue associated with the AG.

Also upregulated in the AG, carbonic anhydrase-regulated protein 10 (*CARP X*) is the only CARP found in protostome invertebrates [48]. CARP X is a catalytically inactive isoform of carbonic anhydrase (CA), which is known to have an osmoregulatory function in the gills of *C. quadricarinatus* [49]. CARP X has also been shown to have a synaptic function, being a secreted glycoprotein ligand of neurexin, which is important for the formation of neuromuscular synapses [50]. It is not clear what *CARP* X role has in the AG function.

A sarcoplasmic calcium-binding protein (*SCP*), associated with moult stages in *P. clarkii*, was upregulated in the *Cq-IAG*-silenced AG. *SCP* is expressed in muscle tissue in *P. clarkii*, primarily in the axial abdominal muscle [51]. *SCP*’s highest expression is during the intermoult stage, and lowest during pre-moult and post-moult [51]. Our investigation showed *SCP* upregulation in the AG tissue, which also included associated muscular sperm duct tissue. As indicated by the moult records and gastrolith harvests from the investigation, in both groups, most individuals were intermoult. If the groups were in different moult stages, it would also be expected for other moult-related genes to be differentially expressed, which was not the case. It is therefore possible that *SCP* has a different function in this species, or a different function in the sperm duct muscle compared to the axial abdominal muscle.

### 3.3. Differential Expression in the Testes

#### 3.3.1. Upregulated Transcripts

Exploring the cascading effects further down the reproductive axis in the testis reveals more than twice the number of differentially expressed transcripts with a far more even split between upregulated and downregulated transcripts. In total, 63 transcripts were differentially expressed in the testis, 34 of which were upregulated. The functions of the upregulated transcripts primarily involved stress, cell growth, and neural response.

Glutathione peroxidase 3 (*GPx3*), a stress-responsive antioxidant upregulated in the testis tissue, was recently characterised in *C. quadricarinatus*, demonstrating upregulation in response to environmental stressors such as high temperature and hypoxia [52]. GPx3 reduces oxidative stress by catalysing the reduction in intracellular hydrogen peroxide to oxygen and water [53]. Two GPx3 cDNA isoforms have been previously characterised: *CqGPx3a*, expressed primarily in the nervous tissue; and *CqGPx3b*, expressed primarily in the walking legs [52]. *CqGPx3a* was upregulated in response to environmental stressors while *CqGPx3b* was unaffected, and *CqGPx3b* was suggested to function as an antimicrobial peptide [52]. Despite clear tissue specificity in the aforementioned study, both *CqGPx3b* and *CqGPx3b* precursors were characterised as the same upregulated transcript due to high homology. Increased expression of *GPx3* in response to *Cq-IAG* silencing could indicate higher stress in the experimental group as a result of cell degeneration observed in the testis tissue. Marine scallops *Chlamys farreri* and *Patinopecten yessoensis* have been shown to upregulate GPx3 in response to exposure to toxic dinoflagellates, which may be due to a stress or antimicrobial response [54]. In human muscle cells, inhibition of GPx3 has been linked to insulin resistance, pathogen vulnerability and inflammatory response [55], providing a loose link with the insulin pathway.

Additionally, a discs large homolog 5 (*DLG5*)-like transcript, associated with cell proliferation, was upregulated in the testis of the *Cq-IAG*-silenced group. Highly conserved between vertebrates and invertebrates, DLG5 is a scaffold protein with multiple functions, including cell growth, polarity, and adhesion. Knockdown of DLG5 in *Drosophila melanogaster* has been shown to cause a reduction in wing growth targeting cell proliferation [56]. In the testis histology images (Figure 5), it can be seen that there are many spermatogonia in the *Cq-IAG*-silenced group, but they do not appear to be progressing to mature spermatozoa as seen in the healthy control testis. Upregulation of *DLG5* could indicate an increase in the cell proliferation rate, but a reduction in the rate of sperm maturation. Although the precise function of these proteins is unclear, these results confirm abnormal testis functioning in the absence of IAG, indicating that sperm maturation is reliant on the presence of IAG.

*PELPK1*, also upregulated, refers to a unique pentapeptide motif that forms a subgroup of the hydroxyproline-rich glycoproteins (HRGP) with PELPK2. It is known to be involved in regulating cell growth and development in the angiosperm *Arabidopsis thaliana*; however, its function in crustaceans has not been investigated [57]. Largely unstudied in animals, PELPK1-like transcripts have been identified but not explored in several decapod species, including *P. clarkii* (XP_045596490.1) and the American Lobster *Homarus americanus* (XP_042219956.1). Their function within the reproductive axis is not known, but their plausible conserved role in the regulation of cell growth should be investigated.

Two cytoskeletal transcripts were upregulated in the testis, those being *actin* and myosin regulatory light chain 2 (*MYL2*). In the noble crayfish *Astacus astacus, actin* has been found to be downregulated in seven-day post-mating spermatophores compared to freshly ejaculated spermatophores; however, the reason for this was unknown [58]. *Actin* could have a role in the cytoskeletal formation of proliferative cells given the higher proportion of spermatogonia cells in the *Cq-IAG-*silenced group. Another upregulated muscle protein *MYL2* is part of the calcium-binding protein superfamily and is relevant to phosphorylation-driven pathways in muscle tissue [59]. This may have a higher expression in the *Cq-IAG*-silenced group due to proportionally more muscle being present in the sample, as there was little sperm produced. The involvement of *MYL2* in protein phosphorylation, potentially converging with the TKIR phosphorylation cascade [40,41], could be investigated using a phosphoproteomic approach.

A sine oculis homeobox protein (*SIX1*) involved in tissue development and differentiation was also upregulated in the *Cq-IAG*-silenced testis tissue [60]. SIX1 is a transcription factor highly conserved throughout vertebrate and invertebrate groups [61,62]. Overexpression of *SIX1* has been noted in various tumours and it has been suggested that upregulation may inhibit apoptosis [60]. The upregulation of *SIX1* in the testis tissue where necrosis was observed may indicate some form of tumorigenesis, or unregulated cell proliferation given the much higher proportion of spermatogonia in the testis. Further studies would be required to confirm the reason for the upregulation of *SIX1* in response to *Cq-IAG* silencing.

Several upregulated transcripts in the testis were found to be related to neural functions. Two separate N-acetylated-alpha-linked acidic dipeptidase 2-like proteins (*NAALAD2*) were annotated, with both identified as having domains related to either glutamate hydrolysis or peptidase activity. NAALAD2 is known to be broadly expressed in ovary and testis tissue as well as brain tissues and is thought to be a glutamate-related neurotransmitter in the central and peripheral nervous systems [63]. Additionally, a forked end (fend) transmembrane protein was upregulated in the testis, known to be an axon guidance molecule, which has a role in neuromuscular recognition and specificity [64]. The upregulation of neurotransmitters and an axon guidance molecule may indicate neural signalling, potentially involved in behavioural changes associated with sexual redifferentiation.

#### 3.3.2. Downregulated Transcripts

A further 29 transcripts were downregulated in the testes in response to *CqIAG*-silencing. Of the 21 that were annotated, the most notable functions were in preventing cell apoptosis or necrosis. An *A-kinase anchor protein* (*AKAP*) was found to be downregulated in the *Cq-IAG*-silenced group. AKAPs are regulatory subunits for protein kinase A (PKA), also known as cAMP-dependent protein kinase, which functions in several signalling cascades involving cell growth, glycogen regulation, and lipid metabolism [65]. It has been previously demonstrated that cAMP is required for cell viability and can inhibit apoptosis [66]. Downregulation of this transcript could be involved in the necrosis of testis tissue observed across several of the *Cq-IAG*-silenced animals. A *zinc finger 99*-like protein with multiple C2H2 domains was also downregulated in the testis. Zinc finger proteins are diverse transcription factors; however, among their roles is the regulation of apoptosis [67]. They also have a key role in tissue development and differentiation [68]. Specifically, the C2H2 domains of zinc finger proteins are involved in apoptosis regulation, cancer growth, and tumorigenesis [67,68]. The presence of multiple C2H2 domains in this protein implies the role of *zinc finger 99* in the degradation of testis tissue. Both *AKAP* and *zinc finger 99* (as well as homeobox protein *SIX1* discussed previously) appear to be involved in the regulation of apoptosis in the testis tissue. The necrosis observed in the *Cq-IAG-*silenced testis tissue (Figure 5) is likely a result of the abnormal expression of these transcripts.

*Giantin*, also known as golgin subfamily B member 1, was also downregulated in the *Cq-IAG-*silenced testis. As its name suggests, *giantin* is a very large (376 kD) protein involved in vesicle trafficking in the Golgi complex [69]. Multiple coiled coil domains were identified in this transcript, which are thought to be involved in the formation of intermolecular complexes or the intercisternal linking and transport of these vesicles [70]. *Giantin* may also be involved in the structure and maintenance of the Golgi complex [70]. Downregulation of *giantin* could indicate the reduced function of the Golgi complex, potentially related to the lack of cell differentiation and transmission.

A protein from bacteria *Vibrio parahaemolytics* and periaxin from fungal pathogen *Colletotrichum tanaceti* were also identified in both groups; however, they were significantly higher in the control group in comparison to the *Cq-IAG-*silenced group. The presence of bacteria in these tissues could indicate infection or contamination; however, aside from the described phenotypic abnormalities in the gonads, all animals in both groups appeared healthy.

## 4. Conclusions

In the seven decades of research since the AG was first discovered, many elements of sexual development in decapods have been elucidated; however, there are many aspects still to be explored. The findings of this investigation aim to further the understanding of the pathways of sexual differentiation and extend our knowledge of the proteins, genes, and receptors within the reproductive neuroendocrine network. The most notable effect of *Cq-IAG* silencing in adult intersex *C. quadricarinatus* was an effective sexual shift, with the testes being functionally arrested and the ovary becoming active in individuals who were previously functional males. Through the construction and analysis of a transcriptomic library, it has been demonstrated that the regulation of the male reproductive axis likely occurs post-transcriptionally, given that the factors known to be involved in this signalling system were not differentially expressed. Feedback between the gonads and the regulatory XO-SG complex in the eyestalk also most likely occurs via a similar post-transcriptional mechanism. Several factors related to cell proliferation, stress, cell repair, apoptosis, and signalling pathways were differentially expressed in the AG and testis tissues, indicating cascading effects throughout the reproductive system on a transcriptomic level. Being of great interest to many stakeholders and having extensive applications towards both aquaculture and conservation management, there is merit in the continued research towards the mechanisms of sexual differentiation, as well as the broader scope of neuroendocrine networks in crustaceans.

## 5. Materials and Methods

### 5.1. Gene Silencing and Dissection

Adult intersex *C. quadricarinatus* (20–50 g) were collected from Freshwater Australian Crayfish Traders (FACT), Tarome, Queensland. Animals were screened onsite and identified as intersex by their gonopore configuration; that being the possession of diagonal male and female gonopores at the base of the fifth and third walking legs, respectively (Figure 8). Animals were maintained in the aquaculture facilities at the University of the Sunshine Coast in 200 L tanks at 24 ± 2 °C and were fed twice weekly with commercially available tropical fish pellets consisting of 36% protein and 4% fat (Aquamunch, Maroochydore, Queensland, Australia). The animals were acclimatised for three weeks, then a control group (n = 6) and the experimental group (n = 6) were selected based on suitable gonopore configuration (Figure 8). A pleopod was removed from each animal prior to injections starting and regeneration after moult was monitored throughout the experiment for phenotypic changes. The pleopods are secondary sexual characteristics related to maternal care in females, providing a clear external morphological distinction between masculinised and feminised individuals [31].

Gene knock-down using RNAi was used to manipulate specific gene expression. Animals were injected with 5 µg double-stranded RNA per gram of body mass at the junction of the tail muscle at the base of the fifth walking leg once weekly over thirteen weeks (5 µg/µL dsRNA synthesised by Genolution, South Korea, via proprietary in-vitro transcription platform). The control group was injected with ds*GFP*, which is known to have no effect on the reproductive system. The experimental group was injected with ds*Cq-IAG*, which has been used to silence *Cq-IAG* in previous experiments (Rosen et al., 2010). The dsRNA sequences correspond with the coding region of the genes used (GenBank accession numbers: *GFP*-ABE28520.1; *Cq-IAG*-ABH07705.1; see Appendix A).

After thirteen weeks of injections, animals were culled in ice slurry and dissected. Tissues from the ovary, testis, sperm duct, AG, and eyestalk were harvested, documented using a dissecting microscope, and weighed where feasible. Gonadosomatic indices were calculated, and the percentage of testes and ovary mass of each specimen’s body weight was calculated in order to control for the differences in specimen size. Analysis of data was conducted in R 4.1.1 [71]. The data were tested to see if they fit the assumptions of homogeneity of variance and heteroscedasticity, using a Shapiro–Wilk test and a Levene’s test, respectively. Upon evaluation, the ovary index data showed left-skewedness and was cube-root transformed to normalise the data. The difference between the treatment and control groups for each response variable was compared via a one-way Analysis of Variance (ANOVA), with treatment as a fixed factor and the %testes mass or %ovary mass as the responses.

For histological analysis, tissues were harvested from the testis, AG, and sperm duct of *Cq-IAG*-silenced animals. Tissues were fixed in Bouin’s solution (Sigma, Melbourne, Victoria, Australia), then dehydrated in sequentially concentrated ethanol from 50% to 100%, soaking for one hour at each concentration, followed by an hour wash with xylene. Tissues were then placed in melted paraffin for one hour and then embedded within cassettes, then sectioned to 5 μm using an automatic microtome (Leica, Melbourne, Victoria, Australia). Slides were washed in xylene to remove paraffin, and then stained with haematoxylin and eosin (H&E) following conventional procedures [72]. Images were taken using a compound light microscope. Previously obtained images of untreated male reproductive tissues were used as controls.

### 5.2. RNA Extraction and Transcriptome Analysis

RNA was extracted from the testis, eyestalk, and AG (with associated proximal sperm duct) for four individuals from each group using RNAZol (Sigma-Aldrich) following the manufacturer’s protocol. RNA was quantified using NanoDrop 2000 (ThermoFisher, Australia). Extracted RNA for the total 24 samples (3–5 µg each) was desiccated with RNAstable (Sigma-Aldrich) and sent to Novogene (Hong Kong) for quality control, followed by library preparation (TrueSeq, with PolyA selection) and RNA Sequencing using HiSeq2500 platform with paired end 150 bp (PE150) sequencing, generating at least 6 Gb fastq files (at least 37 million reads) for each of the 24 libraries.

CLC Genomics Workbench 8.0.3 (CLC; Qiagen, Melbourne, Victoria, Australia) was used to trim reads based on read quality (retaining nucleotides with a minimum Phred score of 20), removing 15 nucleotides from the 3′ end and 5 nucleotides from the 5′ end. Six RNA-Seq libraries, one control and one treated for each of the three tissues, were randomly selected to be de novo assembled using the CLC Genomics Workbench using a minimum contig length of 300 bp. Transcript expressions in each of the 24 samples were quantified relative to library size to map reads per kilobase per million reads (RPKM) data into a dataset for each tissue. A principal component analysis (PCA) was conducted on the top 500 expressed transcripts in each tissue using ClustVis.

Differential expression of transcripts between the treatment and control group was filtered using the parameters: fold change (absolute value) >5, FDR p value correction <0.05, and Bonferroni p value <0.05 (see Appendix A). The transcripts were also filtered using fold change >2 yielding similar results, with no transcripts differentially expressed in the eyestalk regardless of the parameters used. Differentially expressed transcripts were characterised using the TransDecoder function in Galaxy Australia to convert the nucleotide sequences to amino acid sequences, using a minimum amino acid length of 100. BlastP using the nucleotide collection database (nr/nt) on NCBI was then used to characterise these proteins (blast.ncbi.nlm.nih.gov, accessed on 4 October 2022; see Appendix A). The Simple Modular Architecture Research Tool (SMART) was used to identify domains within the protein sequences. The expression of characterised differentially expressed transcripts across tissues with a minimum e-value of 1 × 10^−20^ was visualised with a heat map using Heatmapper (heatmapper.ca, accessed on 4 October 2022). The known sequences of the genes of interest were used as a query in a BLAST search in CLC genomics workbench to identify homologies in the assembly. Once identified, these transcripts were filtered for differential expression using the same parameters listed previously.

## Figures and Tables

**Figure 1 ijms-24-03292-f001:**
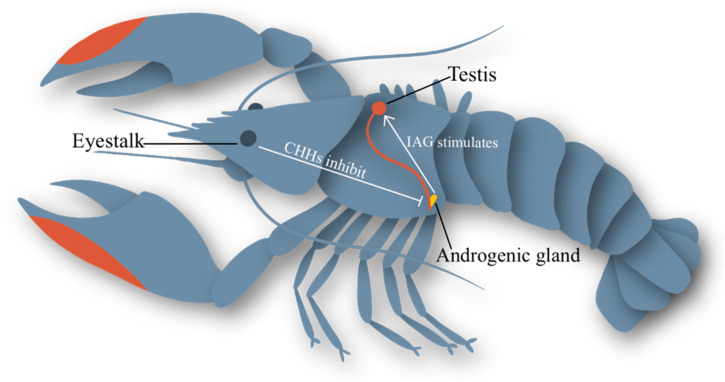
The IAG endocrine network. Inhibitory CHH peptides produced in the eyestalk negatively regulate the function of the androgenic gland, located at the base of the sperm duct at the fifth walking leg. IAG, produced by the androgenic gland, stimulates sperm production in the testes. Feedback loops at all stages of the reproductive axis are not understood.

**Figure 2 ijms-24-03292-f002:**
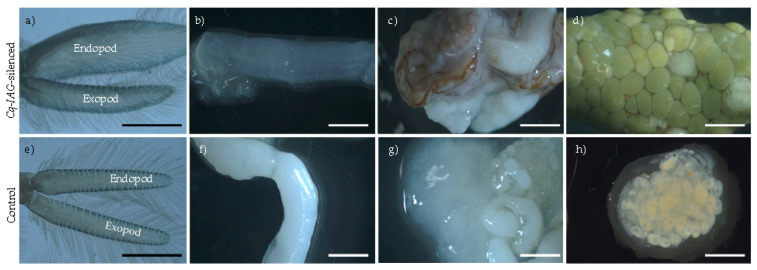
Morphoanatomical features of *Cq-IAG*-silenced crayfish (**a**–**d**) compared to control crayfish (**e**–**h**). Tissues are pleopod (**a**,**e**); sperm duct (**b**,**f**); testis with convoluted sperm duct (**c**,**d**); and ovary (**d**,**h**). In comparison with the control regenerated pleopod (**e**), the regenerated pleopod from *Cq-IAG*-silenced group (**a**) has clearly been feminised. The endopod is twice the width of the endopod, as typical in mature females, whereas the endopod and exopod of the control pleopod are the same size, typical in males. The *Cq-IAG*-silenced sperm duct (**b**) is translucent and entirely devoid of sperm, compared to the control sperm duct (**f**), which appears active. Necrosis was observed in the testis and sperm ducts of *Cq-IAG*-silenced animals (**c**), while all animals in the control group appeared to have normal functioning testes (**g**). The ovaries of the control intersex group were small and pre-vitellogenic (**h**), whereas the ovaries of *Cq-IAG*-silenced individuals were actively producing yolk with oocytes enlarged and vitellogenic (**d**), as would be expected in reproductively active females. All scale bars are 3 mm.

**Figure 3 ijms-24-03292-f003:**
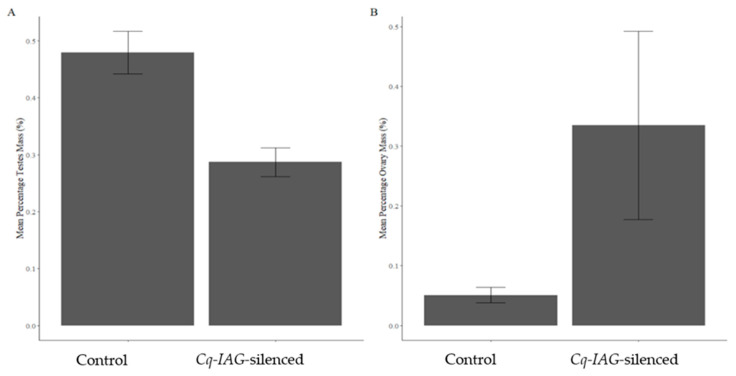
Comparison of gonadosomatic index in control and *Cq-IAG*-silenced groups across ovary (**A**) and testis (**B**) tissues. In comparison to the control group, the *Cq-IAG-*silenced group shower significantly lower testis mass (F (df residual = 9, df test = 1) = 16, *p* = 0.00311) with a mean %mass of 0.287 ± 0.0275 compared to the control groups mean %mass 0.479 ± 0.0408. The *Cq-IAG-*silenced group shower significantly greater ovary mass (F (df residual = 9, df test = 1) = 6.7, *p* = 0.0292) with a mean %mass of 0.335 ± 0.157 compared to the control groups mean %mass 0.0511 ± 0.0129.

**Figure 4 ijms-24-03292-f004:**
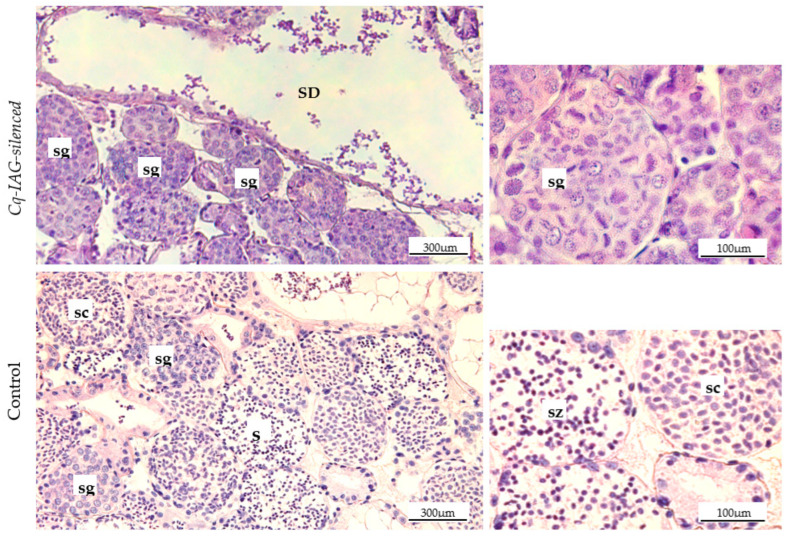
Haematoxylin and eosin-stained cross section from IAG-silenced testis and untreated control testis. In *Cq-IAG*-silenced tissues, undifferentiated spermatogonia (sg) are abundant and dense within the tissue. No mature cells are present, and the testicular lobules appear to be arrested prior to secondary meiosis, evidenced by the lack of cells with condensed cytoplasm. The sperm duct (SD) is devoid of sperm fluid or sperm packets, with only remnants of degenerated cells visible. In the control testis, cells in different phases of maturity can be seen in adjacent lobules. Spermatogonia are less dense, and spermatocytes (sc) and mature spermatozoa (sz) common. The tissue appears active and healthy.

**Figure 5 ijms-24-03292-f005:**
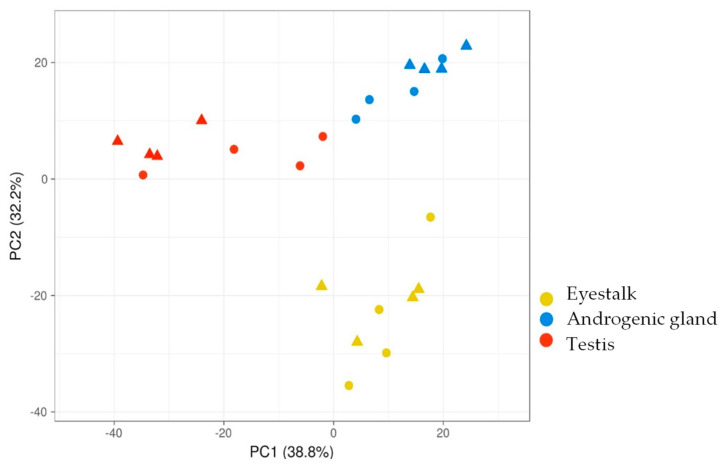
Principal component analysis of the top 500 expressed genes across the 24 libraries derived from three tissues. X and Y axis show principal component 1 and principal component 2, which explains 38.8% and 32.2% of the total variance, respectively. N = 24 data points. There is no notable grouping of treatment groups. Triangles indicate the control group and circles indicate the experimental group.

**Figure 6 ijms-24-03292-f006:**
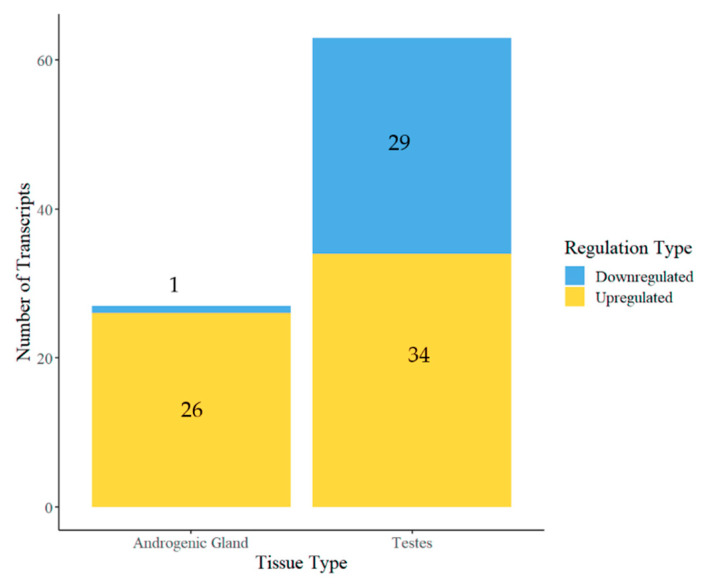
Differentially expressed transcripts in the androgenic gland and testis following CqIAG silencing. There were 27 and 63 transcripts that were differentially expressed in the androgenic gland and testis, respectively, for a total of 90 differentially expressed transcripts. In the testis, 29 transcripts were downregulated and 34 transcripts were upregulated in the *CqIAG*-silenced group compared to the control group, whereas in the AG, one transcript was downregulated (*Cq-IAG*) and 26 transcripts were upregulated in the *Cq-IAG*-silenced group compared to the control group.

**Figure 7 ijms-24-03292-f007:**
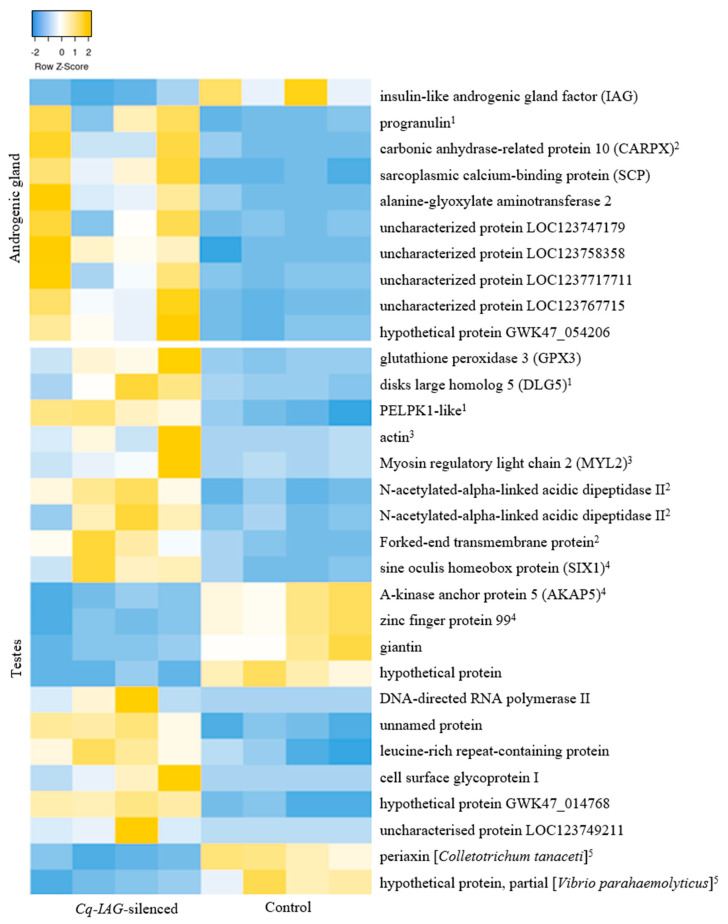
Heat map of differential transcript expression between the *Cq-IAG*-silenced and control groups across the testis and androgenic gland tissues. Predicted functions indicated in superscript are 1—cell growth; 2—neural; 3—muscle; 4—apoptosis/necrosis; and 5—contamination. Yellow indicates a higher RPKM and blue indicates a lower RPKM.

**Figure 8 ijms-24-03292-f008:**
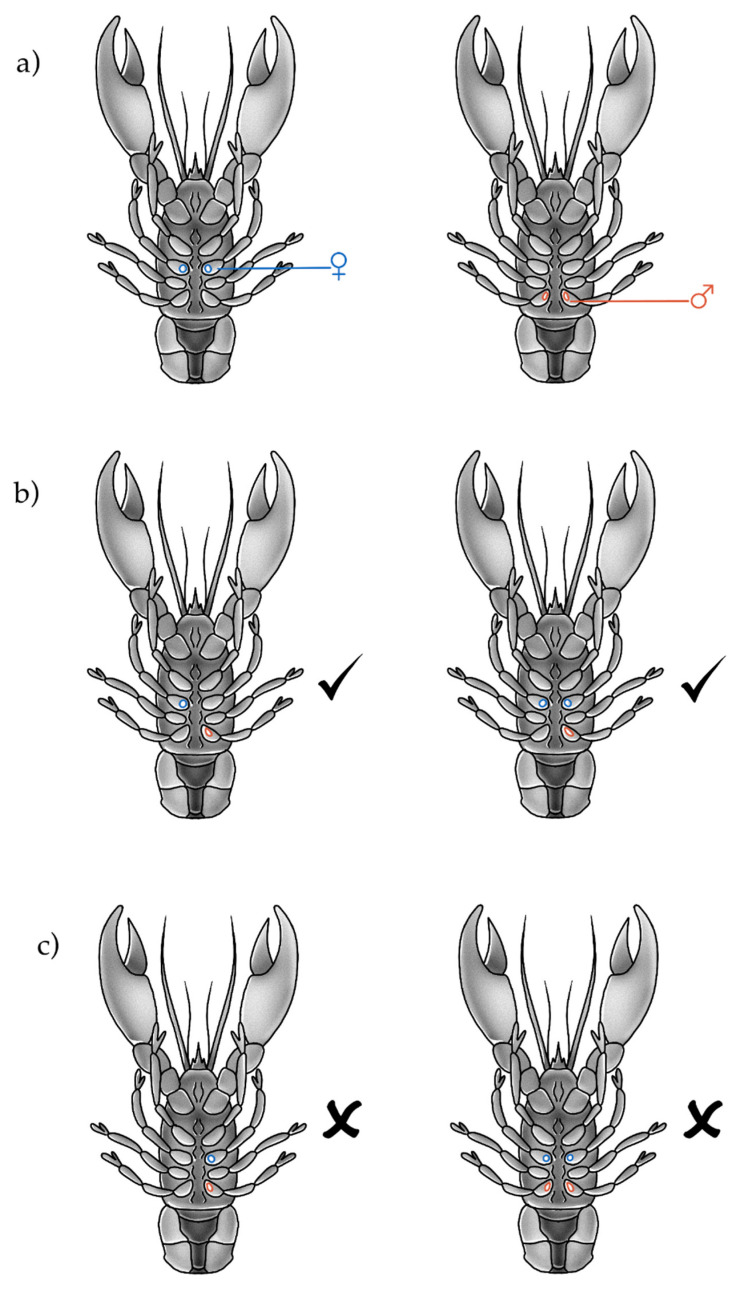
Possible gonopore configurations for candidate crayfish: (**a**) Regular female (left) and male (right) gonopore configuration. Females possess two gonopores at the base of the third walking leg (indicated in blue), while males possess two gonopores at the base of the fifth walking leg (indicated in red). (**b**) A crayfish possessing both male and female gonopores is intersex. To possess internal non-paired male and female reproductive systems as required by the experimental design, an intersex crayfish must have one male gonopore and a female gonopore on the opposite side (left) or both sides (right). (**c**) When two gonopores are present on the same side, the tissue on that side will be masculine. For this reason, individuals with male and female gonopores on the same side only (left) or both sides (right) were not used in this investigation.

## Data Availability

The 24 tissue FASTQ files are available through NCBI Sequence Read Archive under accession number PRJNA862554 and relative expression data for all 24 samples are available through CrustyBase.

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
