# Peer review of "Transcriptomic Changes Following Induced De-Masculinisation of Australian Red Claw Crayfish Cherax quadricarinatus"

_ijms, 2023, doi:10.3390/ijms24043292_

Round 1

Reviewer 1 Report

Some word spelling check required and can add abbreviation as special note. 

Author Response

Spelling throughout has been checked and minor changes have been made. We have not found any additional terms that need abbreviating.

Reviewer 2 Report

Very nicely written and very informative paper!

The major critisism: 

No differential expression was detected between the control and exerimental groups in the eyestalk, however fold change >5 that was used as a cut off (from materilas and metods , lines 706-707) is a very stringent cut off . Even though this cut off produced decent numbers of differentially expressed genes in testes and AG, I would recommend to analyze the data using a less stringent cut off of 2 fold  or even without fold change cut off (just by FDR and/or Bonferroni p values < 0.05) to detect small but statistically significant changes. Less stingent cut offs (typically 2-fold and p<0.05) are used in most gene expression studies. Such analysis might lead to additional findings especially in eyestalk, and might lead to different conclusions.

The paper can be drammatically improved by validating the RNA-seq data with orthogonal methods such as qPCR but especially measuring the protein levels (western blot, etc.) since post-transcriptional regulation is proposed in discussion and in conclusions to explain the results.

Minor points:

lines 222-223 "relatively little genomic and transcriptomic analysis ... on C. quadricarinatus..." - please list the references and discuss the main findings. In discussion - compare the results obtained here with what was found before or comment on why not comparable.

Line 302. 2.2 should be called "RNA-seq results"

Line 331 "MAG was too fragmented to analyze" - please clarify what you mean.

Line 404. Please add the cut off parameters for differential expression and comment on how this changes with less stringent cut offs (2-fold instead of 5-fold and with no fold change cut off - just by Pvalue of < 0.05).

Line 424, generally, I would recommend not to emphasize the numbers of differentially expressed genes since they depend on the cut offs. Please indicate the results of analysis with less stringent cut offs (see previous comment to line 404).

Line 685 "RNA was isolated and extracted" - redundency or please clarify the difference between isolated and extracted, if organs were isolated and RNA was extracted?

Lines 690-693 Please provide details; total RNA input, exact library prep protocol (polyA selection-?), how many reads per sample was achieved.

line 694-696. Please clarify why nucleotides were trimmed.

Lines 706-707. Please report or comment on analysis with 2-fold cut off and/or with no fold change cut-off (just by p< 0.05)

Author Response

Reviewer 2:

Very nicely written and very informative paper!

The major critisism:

No differential expression was detected between the control and exerimental groups in the eyestalk, however fold change >5 that was used as a cut off (from materilas and metods , lines 706-707) is a very stringent cut off . Even though this cut off produced decent numbers of differentially expressed genes in testes and AG, I would recommend to analyze the data using a less stringent cut off of 2 fold  or even without fold change cut off (just by FDR and/or Bonferroni p values < 0.05) to detect small but statistically significant changes. Less stingent cut offs (typically 2-fold and p<0.05) are used in most gene expression studies. Such analysis might lead to additional findings especially in eyestalk, and might lead to different conclusions.

Response: At a less stringent fold change cut-offs, >2 and >0, no transcripts were differentially expressed in the eyestalk. There does not appear to be feedback to the eyestalk on a transcriptional level, even minutely. This was addressed in the revised manuscript in lines 710-713:

“The transcripts were also filtered using fold change >2 and no fold change cut off (just p<0.05) yielding similar results, with no transcripts differentially expressed in the eyestalk regardless of the parameters used.”

The paper can be dramatically improved by validating the RNA-seq data with orthogonal methods such as qPCR but especially measuring the protein levels (western blot, etc.) since post-transcriptional regulation is proposed in discussion and in conclusions to explain the results.

Response: We posit that qPCR validation and measuring protein levels of the differentially expressed genes is beyond the scope of the current manuscript. We did not validate Cq-IAG using qPCR as this is a well-established gene sequence and we had a 100% sequence match with the NCBI database when using blastn. We are interested in exploring these genes and their products through orthogonal methods in future.

Minor points:

lines 222-223 "relatively little genomic and transcriptomic analysis ... on C. quadricarinatus..." - please list the references and discuss the main findings. In discussion - compare the results obtained here with what was found before or comment on why not comparable.

Response: Added reference to reviews transcriptomic research in other commercial decapods M. rosenbergii and P. clarkii. Transcriptomic work in C. quadricarinatus involving sexual differentiation and the reproductive axis is summarised in the introduction. Work involving other pathways and functions, such as osmoregulation or viruses, is not relevant to this  manuscript, making the results of this study non-comparable to other studies outside those already discussed.

Line 302. 2.2 should be called "RNA-seq results"

Response: changed heading.

Line 331 "MAG was too fragmented to analyze" - please clarify what you mean.

Response: changed to “MAG could not be identified in the assembly”

Line 404. Please add the cut off parameters for differential expression and comment on how this changes with less stringent cut offs (2-fold instead of 5-fold and with no fold change cut off - just by Pvalue of < 0.05).

Response: This is addressed later in the text as using a less stringent cut off had the same results (Lines 713-716).

Line 424, generally, I would recommend not to emphasize the numbers of differentially expressed genes since they depend on the cut offs. Please indicate the results of analysis with less stringent cut offs (see previous comment to line 404).

Response: This is a good point. The sentence has been changed to indicate the fold change used to obtain these numbers. We did not feel that discussing the results of less stringent cut-offs here would benefit the reader. Less stringent cut-offs are discussed lines 713-716.

Line 685 "RNA was isolated and extracted" - redundency or please clarify the difference between isolated and extracted, if organs were isolated and RNA was extracted?

Response: removed “isolated” from sentence.

Lines 690-693 Please provide details; total RNA input, exact library prep protocol (polyA selection-?), how many reads per sample was achieved.

Response: the total RNA input was 3-5 µg from each RNA sample (added in line 691). The TruSeq kit uses PolyT beads to select for mRNA from total RNA samples (added in lines 693-694). At least 37 million reads were retrieved per library (added to lines 695-696).

line 694-696. Please clarify why nucleotides were trimmed.

Response: The cutoff used is Phred score 20 (added in lines 698-699).

Lines 706-707. Please report or comment on analysis with 2-fold cut off and/or with no fold change cut-off (just by p< 0.05)

Response: Method and results of fold change >2 and no fold change cut off added in lines 713-716 as follows:

“The transcripts were also filtered using fold change >2 yielding similar results, with no transcripts differentially expressed in the eyestalk regardless of the parameters used.”

Reviewer 3 Report

It is very nice paper. Amazing photos to illustrate some topics within the manuscript.

Comments:

My mean concern is the incomplete description of the RNAi sequences in Methods section. Complete this information must be mandatory to approve the manuscript.

Only the genes are mentioned, however, to reproduce this assay it is necessary to complete the information. The inclusion of a table with the following information will overcome this issue: size of the dsRNAs injected; only 1 designed dsRNA per gene? if more than 1, so provide each size and sequence; describe the method of preparation dsRNA, (expressed in E.coli?; synthetic or in vitro synthesis?), it was necessary a method for concentration ?.

Figure 7: The yellow scale it seems to require more definition, maybe a color change would improve the visualization of the results.

Figure 8: The same as above. Positions of the gonospore could be colored to improve visualization.

Author Response

Major comments:

Size of the dsRNAs injected; only 1 designed dsRNA per gene? if more than 1, so provide each size and sequence; describe the method of preparation dsRNA, (expressed in E.coli?; synthetic or in vitro synthesis?), it was necessary a method for concentration ?.

Response: One dsRNA was used per gene (IAG and GFP). These were synthesised by Genolution, South Korea (Line 642). The sequences for these genes are available via the GenBank accession numbers in text (lines 649-650). Concentration used was 5µg/µl and the size is 500 nt per dsRNA sequence (added in line 640).

Minor comments:

Line

Comment

Response

Fig. 7

The yellow scale it seems to require more definition, maybe a color change would improve the visualization of the results

We have changed the colour to darker yellow.

Fig 8.

The same as above. Positions of the gonopore could be colored to improve visualization.

Agreed and changed as suggested.

Round 2

Reviewer 2 Report

1.

lines 710-714 

 in the response that authors state that the phrase now is in the revised text:

“The transcripts were also filtered using fold change >2 and no fold change cut off (just p<0.05) yielding similar results, with no transcripts differentially expressed in the eyestalk regardless of the parameters used.”

However the actual phrase in the text misses " and no fold change cut off (just p<0.05)  " - did this analysis also produced no differentially experessed transcripts?

In the text:

"Differential expression of transcripts between the treatment and control group was filtered using the parameters: fold change (absolute value) >5, FDR p value correction <0.05; and Bonferroni p value <0.05 (see supplementary file 1). The transcripts were also filtered using fold change >2 yielding similar results, with no transcripts differentially expressed in the eyestalk regardless of the parameters used."

2. I disagree that the qPCR validation (or other orthogonal validation) is beyond the scope of this paper. If the authors still have the samples such validation would be highly recommended, it should not be too combersome and significantly increase the soundness of the data to the audience. I would like to leave this to the editors discretion.

Author Response

The fold change 2 showed no change, and there were no additional differentially-expressed genes represented when using this threshold.

We are unable to perform qPCR analysis, as samples were depleted and the project funding is no longer available.

Reviewer 3 Report

Some of my comments were misunderstood. So I keep it for the revised version, I would like see the corrections.

Major comments:

1.- Size of the dsRNAs injected;  Response: Concentration used was 5µg/µl and the size is 500 nt per dsRNA sequence (added in line 640). I am OK with that answer.

2.- only 1 designed dsRNA per gene? if more than 1, Response:  One dsRNA was used per gene (IAG and GFP). I am OK with that answer.

3.- so provide each size and sequence; Response: The sequences for these genes are available via the GenBank accession numbers in text (lines 649-650). I am NOT OK, because I am asking about the sequence of the 500 nt used for dsRNA, not about the sequence of the entire gene.

Is this sequence (about 530) bp the dsRNA used in the experiment? Please provide the right sequence in a table for both dsRNA.

>DQ851163.1:556-1086 Cherax quadricarinatus insulin-like androgenic gland factor mRNA, complete cds ATGCTGTTCCAAACATTACTCAACCTGATTTTGGTTGTGGTGGTGAAGCTGCCTCCTCCCTCCGCCTCTT ACAGAGTGGAAAACCTTCTGATTGACTTCGACTGTGGCCACCTGGCGGACACAATGGACAGTATTTGCCG CACCTACCAGGAATTTAACGACACCCGAGCGGTGAGGTCGGCCAGAGATGCATCATTTTCTGCCAGTGTC TCCATGTATGACCCCGGGAGTAAGATTGCTGTTCGTCAAGTATACCATCCAAGAGGCAGGAAGTTGGGTG TCAAGTTTACTGTCCCTGATGCCAGGTTGGGTAAGCAGGAGGCGATGACAGTGAGTCGCGAGGCCGCCCA TACGTTTATAAAGACCCAGAACTACAACCGTCGCCGCCGTAACTCAGATACGACAGACAATACAAGCAGC ACTAACGTTTATGATGAGTGTTGCAGCGAGAAAACATTGAAGACCTGCGTCTTCGATGAGATTGCCCAGT ACTGTGAACAGTTGGAGGACGGAATCTACGTCAGTTCTTGA

4.- describe the method of preparation dsRNA, (expressed in E.coli?; synthetic or in vitro synthesis?) Response: These were synthesised by Genolution, South Korea (Line 642). I am NOT OK, because I am asking about the method used for Genolution to obtain the dsRNA. In my opinion this description is mandatory to reproduce results.

5.- it was necessary a method for concentration ?. This point was related to the point. 4, and it is about the preparation of dsRNA, I mean: to reach 5ug/ul, there was some concentration method required?

Author Response

3.- so provide each size and sequence

Response: The sequence for each dsRNA has been annotated and added as an additional supplementary file.

4.- describe the method of preparation dsRNA, (expressed in E.coli?; synthetic or in vitro synthesis?)

Response: The method used by Genolution to obtain the dsRNA was proprietary in-vitro transcription platform. This has been added in text lines 640-641.

5.- it was necessary a method for concentration?

Response: Genolution supplies the dsRNA in concentrations higher than 5ug/ul and therefore do not require any concentration protocol, only dilution.

Round 3

Reviewer 3 Report

Nice work.

Author Response

Thank you!